# Design of a Naturally Dyed and Waterproof Biotechnological Leather from Reconstituted Cellulose

**DOI:** 10.3390/jfb13020049

**Published:** 2022-04-29

**Authors:** Claudio José Galdino da Silva Junior, Julia Didier Pedrosa de Amorim, Alexandre D’Lamare Maia de Medeiros, Anantcha Karla Lafaiete de Holanda Cavalcanti, Helenise Almeida do Nascimento, Mariana Alves Henrique, Leonardo José Costa do Nascimento Maranhão, Glória Maria Vinhas, Késia Karina de Oliveira Souto Silva, Andréa Fernanda de Santana Costa, Leonie Asfora Sarubbo

**Affiliations:** 1Rede Nordeste de Biotecnologia (RENORBIO), Universidade Federal Rural de Pernambuco (UFRPE), Rua Dom Manuel de Medeiros, Dois Irmãos, Recife 52171-900, PE, Brazil; claudiocjg@gmail.com (C.J.G.d.S.J.); julia_amorim@hotmail.com (J.D.P.d.A.); alexandre_dlamare@outlook.com (A.D.M.d.M.); 2Instituto Avançado de Tecnologia e Inovação (IATI), Rua Potyra, n. 31, Prado, Recife 52171-900, PE, Brazil; andrea.santana@ufpe.br; 3Escola Icam Tech, Universidade Católica de Pernambuco (UNICAP), Rua do Príncipe, n. 526, Boa Vista, Recife 52171-900, PE, Brazil; 4Centro de Tecnologia em Design de Moda, Faculdade Senac Pernambuco, Rua do Pombal, n. 57, Santo Amaro, Recife 52171-900, PE, Brazil; anantchalafaiete@gmail.com; 5Centro de Tecnologia e Geociências, Departamento de Engenharia Química, Universidade Federal de Pernambuco (UFPE), Cidade Universitária, s/n, Recife 52171-900, PE, Brazil; helenise_almeida@hotmail.com (H.A.d.N.); mariana.ahenrique@ufpe.br (M.A.H.); gloria.vinhas@ufpe.br (G.M.V.); 6Centro de Tecnologia, Departamento de Engenharia Têxtil, Universidade Federal do Rio Grande do Norte (UFRN), Avenida Senador Salgado Filho, n. 3000, Lagoa Nova, Natal 59078-970, RN, Brazil; leonardo.nascimento.063@ufrn.edu.br (L.J.C.d.N.M.); kesia.souto@ufrn.br (K.K.d.O.S.S.); 7Centro de Comunicação e Design, Centro Acadêmico da Região Agreste, Universidade Federal de Pernambuco (UFPE), BR 104, Km 59, s/n, Nova Caruaru, Caruaru 50670-901, PE, Brazil

**Keywords:** fashion, design, sustainable clothing, bacterial cellulose

## Abstract

Consumerism in fashion involves the excessive consumption of garments in modern capitalist societies due to the expansion of globalisation, especially at the beginning of the 21st Century. The involvement of new designers in the garment industry has assisted in creating a desire for new trends. However, the fast pace of transitions between collections has made fashion increasingly frivolous and capable of generating considerable interest in new products, accompanied by an increase in the discarding of fabrics. Thus, studies have been conducted on developing sustainable textile materials for use in the fashion industry. The aim of the present study was to evaluate the potential of a vegan leather produced with a dyed, waterproof biopolymer made of reconstituted bacterial cellulose (BC). The dying process involved using plant-based natural dyes extracted from *Allium cepa* L., *Punica granatum,* and *Eucalyptus globulus* L. The BC films were then shredded and reconstituted to produce uniform surfaces with a constant thickness of 0.10 cm throughout the entire area. The films were waterproofed using the essential oil from *Melaleuca alternifolia* and wax from *Copernicia prunifera*. The characteristics of the biotechnological vegan leather were analysed using scanning electron microscopy (SEM), thermogravimetric analysis (TGA), flexibility and mechanical tests, as well as the determination of the water contact angle (°) and sorption index (s). The results confirmed that the biomaterial has high tensile strength (maximum: 247.21 ± 16.52 N) and high flexibility; it can be folded more than 100 times at the same point without breaking or cracking. The water contact angle was 83.96°, indicating a small water interaction on the biotextile. The results of the present study demonstrate the potential of BC for the development of novel, durable, vegan, waterproof fashion products.

## 1. Introduction

The exponential increase in the world population has led to an increasing demand for products, which has pressured industries and supply chains to develop cheaper products at a fast pace and on a large scale. Such products generally have low durability, resulting in considerable socioenvironmental harm. The increase in industrial production also contributes to the intensification of pollution. Thus, numerous consumers, researchers, and specialists have become concerned with issues of production and consumption and the sustainable management of the supply chain [1]. Moreover, the awareness of socio-environmental problems, such as climate change, resource scarcity, the exploitation of labour, and the pollution of water resources, has increased expectations on the part of consumers regarding how companies develop their brands and products [2].

The textile industry is the second-largest polluter globally, behind only the oil industry. Thus, textile companies have been looking for safer materials to replace current highly polluting manufacturing methods, and studies on sustainable alternative raw materials are currently being conducted worldwide [3,4,5]. Bacterial cellulose is one of these biomaterials with considerable potential for use in the fashion industry [6].

Although plants are the major source of cellulose, several genera of microorganisms are also capable of producing this substance, which is known as bacterial cellulose or biocellulose (BC). As BC is considered a low-cost, extremely versatile, ecologically correct biopolymer, studies involving different applications of this biomaterial have increased over the years [6,7,8,9,10].

The molecular formula of BC is the same as that of plant cellulose. However, the fibrillar structure of BS is smaller and has a larger surface area. The fibres of plant cellulose have a diameter of approximately 13 to 22 μm, with a crystallinity of 44 to 65% [11,12]. In contrast, BC fibrils are naturally nanometric, with a diameter of 10 to 100 nm and crystallinity close to 90%. Moreover, BC has a high tensile strength (~70 N) and has a hydrophilic nature due to the high number of hydroxyl groups on its surface [12,13].

Due to these characteristics, microbial cellulose can serve as the basis for the sustainable textiles that the fashion industry seeks. The biomaterial can be produced in different shapes and thicknesses, taking on the shape of the recipient in which the microbial fermentation is performed, thereby avoiding waste in the modelling of parts. It is also possible to use small patches of BC to make larger pieces through homogenisation and remodelling of the material. The production of BC can also use agro-industrial waste products, thereby lowering costs and making the product more accessible. Moreover, BC is easily degradable when discarded in nature [3,14,15].

Another factor that can aggregate value to biocellulose in fashion is dyeing using natural pigments, which provide colour, tonality, aesthetics, and sustainability to the products. Such dyes can be extracted from different parts of plants (leaves, flowers, fruits, stems, and roots). Besides being sustainable, natural dyes can be successfully used with BC when the colourisation conditions are controlled. Thus, non-polluting, biodegradable products, such as plant-based dyes and biotextiles, have considerable potential for use as novel biotechnological products that meet the needs of the world market [6,16].

However, as a production process that occurs by natural fermentation, some challenges need to be overcome to apply biocellulose as a textile material. It is necessary to ensure a uniform structure with a constant thickness, attractive textures, adequate strength, fit, comfort, water resistance, and durability, along with the maintenance of attractive aesthetics to create novel products [17].

Considering the concepts, trends, and possibilities of biocellulose and natural dyes for use in the fashion industry, the aim of the present study was to produce a biocellulose-based vegan leather using an innovative shredding and reconstitution process to create a more uniform structure with a constant thickness. The biotextile was then dyed naturally with pigments extracted from plants and waterproofed to ensure its applicability in developing novel textile products for increasingly conscientious consumers.

## 2. Materials and Methods

### 2.1. Microorganisms and Means of Maintenance

Microorganisms in a symbiotic culture of bacteria and yeast (SCOBY), obtained from the culture collection of Nucleus of Resource in Environmental Sciences, Catholic University of Pernambuco, Brazil, were used to produce BC. According to Villarreal-Soto et al. (2018) [18], the microorganisms in the microbiological composition of the consortium include acetic acid bacteria (*Komagataeibacter* sp. and *Acetobacter *sp.), lactic acid bacteria (*Lactococcus* sp. and *Lactobacillus* sp.) and yeasts (*Zygosaccharomyces bailii, Saccharomyces cerevisiae* and *Schizosaccharomyces pombe*). The maintenance medium, called green tea medium, is constituted of 50.00 g/L of sucrose and 1.15 g/L of citric acid, acquired from MERTEC (Brazil), and 10.00 g/L of green tea leaves (*Camellia sinensis*) from *Chá Leão* (Brazil), adjusted to pH 6.

### 2.2. BC Culture Conditions, Purification, and Yield

BC production was performed by transferring 10% (*v*/*v*) of a pre-inoculum containing the microorganisms in the consortium to 2500-mL Schott flasks containing 2000 mL of the green tea medium. Static cultivation was performed at 30 °C for 14 days. The BC was rinsed in running water, and purification was achieved by immersion in a 0.1 M NaOH solution at 70 °C for 1 h. The BC films were then neutralised and weighed, followed by calculating the yield.

### 2.3. Water Retention Capacity (WRC)

WRC is linked to moisture. The analysis of this measure determines the capacity of the biomaterial to adsorb and fix dyes. The BC films were weighed (25 °C, 1 atm) and dried in an oven at 60 °C until reaching a constant weight, indicating the complete removal of water. The *WRC* was then obtained using Equation (1):(1)$$WRC\;\left(\%\right)=\frac{Mean\;of\;wet\;weights-Mean\;of\;dry\;weights}{Mean\;of\;wet\;weights}\;$$

### 2.4. Natural Dye Extraction

An infusion was made at room temperature in a solution composed of 1000 mL of deionised water and 250 mL of 70% ethanol to extract dye from *Eucalyptus globulus L.* (50 g of dry leaves), *Allium cepa L.* (50 g of bulb bark) and *Punica granatum* (50 g of dried fruit peel) obtained at public markets in the city of Recife, Pernambuco, Brazil. After 24 h, the infusions were boiled for 30 min and filtered to remove the vegetable matter. The liquids were then used for dyeing.

### 2.5. Preparation of BC Films for Dyeing

A water solution was prepared with the fixative (potassium alum 99.5%) at 20 g/L. The BC films were submerged in the solution and heated at 90 °C for 30 min under agitation for the penetration of the potassium alum and fixation in the fibres of the films.

### 2.6. Dyeing and Natural Dye Fixation Procedure

The volumes obtained from the dye extracts were kept at 90 °C and used for dyeing 1000 g of BC fibres (wet mass). The films were submerged in the heated extracts for 1 h with light agitation. The films dyed with *Allium cepa L.*, *Punica granatum,* and *Eucalyptus globulus L.* were then rinsed in running water and placed in a fixative bath containing 20 g of NaCl/L in water for 30 min.

### 2.7. Shredding, Reconstitution, and Drying

Each dyed BC film was shredded in wet condition with the aid of an industrial blender at 18,000 rpm for 2 min to form a homogeneous mass. This mass was uniformly distributed on a 20 cm × 20 cm silkscreen for reconstitution, and the fibres were dried at room temperature. This process lasted three to six days and only ended when the total reconstitution of the biotextile was observed. At the end of the process, the biotextile had the appearance of thin coloured leather.

### 2.8. Waterproofing with Essential Oil and Wax

Only products of plant origin were used for the waterproofing process to maintain the vegan nature of the leather. Essential oil from *Melaleuca alternifolia* and wax from *Copernicia prunifera* were chosen for this process, as these substances are hydrophobic plant products and are easy to find at public markets in the city of Recife, Pernambuco, Brazil. Immediately after the completion of the reconstitution and drying processes, the reconstituted BC surfaces received a thin layer (applied with a brush) of a mixture of 50% *w/w* wax and oil previously dissolved and homogenised at 75 °C. The samples were then dried at room temperature for 48 h in a naturally ventilated room.

### 2.9. Characterisation of Biotechnological Vegan Leathers

#### 2.9.1. Determination of Water Contact Angle and Sorption Index

Rectangular samples, with 10 mm in height and 5 mm in length, of dried vegan leather were used for the analysis. To establish the behaviour of the biomaterials in contact with water, each sample was placed in a holder so that the material’s surface was flat. Contact angles were determined with the aid of a goniometer using the sessile drop technique. A digital camera (XT10, Fujifilm, Japan) was used for analysis. A droplet of ~25 μL was placed carefully on the upper surface of the leather, and the contact angle was recorded after 1.0 s of spreading [19]. To determine the sorption index (s)*,* the droplet was observed for 10 min until complete water absorption, and the average time was calculated [20].

#### 2.9.2. Swelling Ratio

Rectangular, with 10 mm in height and 5 mm in length, of dried vegan leather samples were weighed and immersed in a 100-mL distilled water bath at 25 °C for 24 h. The samples were then removed from the bath. The excess water was carefully removed from the leather surface with tissue paper, and the samples were weighed immediately. Swelling ratios were determined from the change in weight before and after swelling and expressed as: (2)$$\;SR\left(\%\right)=\frac{Swollen\;weight-Initial\;weight}{Initial\;weight}\times 100\;$$

#### 2.9.3. Scanning Electron Microscopy (SEM)

For the SEM analysis, dried vegan leather samples were mounted on a copper stub using double adhesive carbon conductive tape and coated with gold for 30 s (SC-701 Quick Coater, Tokyo, Japan). The SEM photographs were obtained using a scanning electron microscope (MIRA3 LM, Tescan, Warrendale, PA, USA) operating at 10.0 kV at room temperature.

#### 2.9.4. Thermogravimetric Analysis (TGA)

The thermal stability of the samples was determined using TGA. Approximately 8 mg of each sample was heated from 30 to 600 °C at a rate of 10 °C/min in a nitrogen atmosphere with a flowrate of 20 mL/min to avoid the oxidative degradation of the samples. The Mettler Toledo TGA 2 Star System was used for this analysis.

#### 2.9.5. Flexibility

To test flexibility, the vegan leather samples were folded by hand 100 times along the same line. The classification of flexibility was based on the number of folds until failure: poor (<20), fair (20–49), good (50–99), and excellent (≥100) [21].

#### 2.9.6. Mechanical Test

Tensile strength (N) and maximum deformation (%) according to time (s) were determined based on Rethwisch and William (2016) [22] for the characterisation of the mechanical properties of the BC. Samples of the dried vegan leathers were cut into rectangular strips (7.5 cm × 3 cm). The mean leather thickness was 0.10 cm. The tensile strength test was performed at room temperature at a velocity of 5 m/min and a static load of 0.5 N using a universal testing machine (EMIC DL–500MF, Brazil), following the ASTM D882 method. 

## 3. Results and Discussion

### 3.1. BC Yield and Water Retention Capacity

The mean yield of the hydrated BC was 422.12 ± 15.26 g de cellulose/L of fermentation medium. Regarding the dried BC films, the mean yield was 10.07 ± 1.97 g/L with a production time of 14 days. This yield is considered satisfactory when compared to yields reported in previous studies with the same 14-day production time, such as 4.56 g/L in the study by Salari et al. [23] and 6.18 g/L in the study by Ul-Islam et al. [24].

The purification step with NaOH favoured an even colour as well as the removal of metabolites and possible residues from the culture medium adhered to the surface of the biocellulose. Figure 1 displays the purified BC film.

The results displayed in Table 1 confirm the high WRC (%) of the BC (>97%). Costa et al. [25] and Nascimento et al. [26] describe similar results. The WRC is a fundamental characteristic of efficiency in the incorporation and fixation of hydrophilic dyes.

### 3.2. BC Dyeing, Dye Fixation, and Waterproofing with Essential Oil and Wax

After the purification step, the membranes were submitted to the dyeing process with the pigments obtained from the plant extracts. The dyed samples were then shredded, followed by reconstitution during the drying process. The dried vegan leather samples were submitted to the waterproofing process with a mixture of essential oil and plant wax. The BC samples used to determine the effectiveness of the methods are listed in Table 2.

The dyeing and fixation process resulted in the different colours for each sample, corresponding to different tonalities of the plant extracts used *in natura* (Figure 2). All experiments resulted in good visual quality and even pigmentation, corresponding to a varied swatch of colours that could be used in fashion products. The successful fixation of the dyes in the fibres is in agreement with results reported in the literature, such as studies by Verma et al. [27] (dyeing with onion), Maulik et al. [28] (dyeing with eucalyptus) and Tian et al. [29] (dyeing with pomegranate).

The waterproofing of the surface of the samples also influenced both the visual and physical aspects (Figure 2b,d,f,h), making the material less opaque and adding shine. Moreover, the samples that underwent this process had a more pleasant texture in terms of softness to the touch, losing the initial characteristics of roughness and dehydration.

Thus, the development of a completely vegan microbial cellulose-based textile material submitted to shredding and reconstitution was successful, generating a fabric similar to leather that was more uniform than the post-fermentation BC, with a constant thickness of 0.10 cm. Moreover, the process did not generate any waste. Even the cellulose remnants that did not have the ideal size and thickness for drying and the creation of pieces could be shredded together to form a mass that could be moulded and dried into the desired shape.

Chan et al. [30] also described the non-generation of waste during the production of BC pieces. The researchers used different approaches and novel cultivation techniques, employing recipients with pre-established dimensions so that the cellulose could grow into the desired shape, thereby avoiding cutting and waste and facilitating the creation of future pieces. The authors proved that the organic material can be reused after its fermentation, shredding, and reconstitution and can be grown into any shape of the desired apparel without cuts and with no generation of waste material.

### 3.3. Water Contact Angle, Swelling Ratio, and Sorption Index

The wettability of a fabric is important to sensorial comfort and is determined by the time required for the fabric to absorb a drop of water. The difference in the contact angle of the water droplet is recorded over time. This time-lapse is denominated by the sorption index (s) [31].

The swelling of the fabric offers the possibility of retaining sweat by the material during its use, which can be beneficial from a practical standpoint [20]. It can also indicate the possibility of the material functioning as a protective moisture barrier. The swelling capacity is expressed as the swelling coefficient and depends on the type and quantity of water retention agents in the fabric. This factor is crucial to enabling the control of the desired properties of the fabric in accordance with its application [20]. Table 3 displays the water contact angle, swelling ratio, and sorption index of the different BC samples.

The obtained water contact angles results (Table 3) are in agreement with studies involving the BC’s surface modification with the aim of increasing the water droplets contact angle and, consequently causing a decrease on the hydrophilicity of the biocellulose [31,32].

The vegetal wax was used as a water-repellent agent. It was responsible for creating a layer on the surface of the samples that waterproofed them. As indicated by the sorption index, even after the 600 s time limit was exceeded, the samples that had the protective layer were not able to absorb the liquid. Another factor that indicates the success in this application was the contact angle change between the surface of the samples and the water droplets, indicating that the additional layer reduced the hydrophilicity of the outer layer of the material. Bashari et al. [33] state that the fine particles of *Copernicia prunifera* wax can be used as a natural moisture-repellent agent, therefore responsible for reducing the surface energy of tissues and producing a nano roughness that makes surfaces repellent to moisture.

By observing the swelling ratio results, it was possible to notice a significant value reduction in all the waterproofed samples when compared to the samples without the waterproofing process. Even after being submerged for 24 h, some samples showed a decrease of more than half of the initial swelling ratio value, thus, confirming that the waterproofing process is effective in reducing the swelling ratio (%).

### 3.4. Scanning Electron Microscopy

SEM was used to investigate the morphology of the surface of the samples before and after shredding, dyeing, and waterproofing. The BR sample (Figure 3c) exhibited a uniform surface, as seen in the BC (Figure 3a), indicating that the reconstitution of the BC was successful. Comparing BR (Figure 3c) to BRA and BRE (Figure 3e,g), the surface of the samples exhibited a new covering, which was related to the dyeing process and pigmentation of the biotechnological vegan leather.

In comparison to the samples before waterproofing (Figure 3a,c,e,g), those after waterproofing (Figure 3b,d,f,h) revealed the emergence of a uniform layer on the surface. This layer is the mixture of the essential oil from *Melaleuca alternifolia* and the wax from *Copernicia prunifera,* indicating that the waterproofing process was successful, giving the surface an impermeable characteristic that would hinder the absorption of water. This new layer is also responsible for the new characteristics of shine and softness discussed above. Moreover, the additives absorbed by the surface of the microbial cellulose could reduce undesired effects related to contact of the material with water. Thus, the dyeing and waterproofing methods constitute a simple, ecologically friendly way to improve the properties of biotextiles.

### 3.5. Thermogravimetric Analysis (TGA)

Figure 4 and Table 4 show that the materials developed have somewhat similar profiles. According to Rathinamoorthy et al. [34], the initial decomposition temperature (T_onset_) is when BC begins to disintegrate. This temperature represents the onset of the breakdown of the thermal stability of the material.

Two main stages in the loss of mass were found in the samples without waterproofing (BC, BR, BRA, and BRE), whereas three stages were found in the samples with waterproofing (BC-W, BR-W, BRA-W, and BRE-W). The first degradation stage in all samples occurred between 80 and 330 °C, with a mean loss of 33.31 ± 7.41% of mass related to the evaporation of the water adsorbed to the biocellulose and that in the composition of the pigment on the surface of the dyed samples.

The second stage occurred at around 270 to 370 °C, leading to a mean final mass of 47.14 ± 5.77%. This loss was probably related to the degradation of the cellulose, with its de-polymerisation, dehydration, and decomposition of glucose units, as well as the subsequent formation of carbon residues [35], as the main pyrolysis stage of cellulose occurs in a temperature range of 300 to 380 °C [36].

The third stage occurred between 390 and 518 °C and only in the samples with wax in their composition (BC-W, BR-W, BRA-W, BRE-W). Thus, this additional stage was likely related to the waterproofing substance in the samples. At the end of this last degradation stage, the mean residual mass was 34.43 ± 10.12%

The results are compatible with findings described in the literature. In one study, cotton fabrics without waterproofing also had two thermogravimetric phases, the first of which was up to 300 °C related to the degradation of the amorphous regions of the cellulose polymer and with a considerable reduction in mass. The second was related to the crystalline regions of the material, with a maximum temperature of 430 °C [36].

Comparing the results of the intact samples (BC and BC-W) and reconstituted samples (BR, BR-W, BRA, BRA-W, BRE, and BRE-W), the degradation temperatures were an average of 42.21 °C lower for the samples without additives, and an average of 53.14 °C lower for the waterproofed samples in the first stage when compared to the T_max_ of the intact samples. This indicates that the shredding and reconstitution of the cellulose exerted a direct impact on the first degradation phase. The behaviour was inverted in the second stage, as the reconstituted sample without additives (BR) had slightly higher T_max_ (349.77 °C) and residual mass (62.05%) compared to the intact BC (339.04 °C; 56.97%). This is an interesting point and suggests that the BR without dyeing and waterproofing exhibits greater thermal stability during the degradation of the cellulose and formation of carbon residues.

The presence of the dye also diminished the thermal stability in the first and second degradation stages, as T_max_ was reduced by 40 °C in the BRA and BRE samples compared to the BR in both stages. Different results were found concerning waterproofed samples. The waterproofing process led to a reduction in T_max_ ranging from 11.69 to 29.26 °C in the first stage, indicating that the degradation of the wax from *Copernicia prunifera* also occurred in this temperature range. According to Pan et al. [37], the thermal decomposition of this wax in its natural form occurs in the range of 270 to 320 °C. Moreover, the degradation of the low molecular-weight components of the essential oil from *Melaleuca alternifolia* occurs around 170 to 240 °C [38]. In the second stage, waterproofing increased the T_max_ of the BRA-W and BRE-W samples by 57.09 °C and 62.80 °C, respectively, indicating a possible interaction between the waterproofing agent and layer of dye, thereby enhancing the thermal stability of these materials.

### 3.6. Flexibility and Mechanical Tests

Flexibility is considered one of the critical aspects of the usability and durability of textile products. Textile materials need a surface structure with enough rigidity for wearability and adequate flexibility to be comfortable [39].

The BC samples exhibited excellent flexibility, remaining intact after being folded by hand more than 100 times at the same point. However, some of the samples with a layer of wax on the surface exhibited cracking of this waterproofing film. Thus, further studies are needed to improve the application of the waterproofing agent.

The functioning of a fabric involves its performance during use and is directly linked to its mechanical properties. The results of the mechanical tests (Figure 5 and Table 5) demonstrated that the addition of the plant-based waterproofing agent was capable of enhancing tensile strength by 3.56 to 30.79% and increasing-albeit subtly-the maximum deformation capacity of all samples. These aspects are beneficial, as the elasticity of textiles is of considerable importance and enables pieces to have specific characteristics, such as lightness, less volume, and a tendency to form fewer wrinkles.

The samples dyed with eucalyptus (BRE and BRE-W) exhibited excellent results regarding maximum deformation. With the addition of the wax, the deformation time of the BRE-W sample was improved by 147.51% compared to the non-waterproofed sample (BRE).

The samples submitted to the reconstitution process also exhibited improved mechanical properties, as demonstrated by comparing the BC and BR results. The shredding and reconstitution of the sample increased maximum deformation and tensile strength nearly doubled, increasing by 81.97 N. This indicates that the reconstitution process not only improves the visual appearance of the leather and standardisation of production, it also enhances the mechanical properties of leather of a biotechnological origin.

As already recognized by researchers, BC is naturally an exclusive combination of properties such as high polymerization degree, high surface area, high flexibility, high tensile strength, and high water-holding capacity [40]. Based on the results, it is safe to assume that the materials proposed have similar or better tensile strength and deformation properties compared to data described in the literature [17,20]. Thus, the findings indicate the successful development of a waterproof textile material made from reconstituted biocellulose.

## 4. Conclusions

The use of biotechnological materials, such as BC, indicates new possibilities for the market of sustainable fashion, which is of interest to both manufacturing companies and conscientious consumers. The growing interest in such materials will contribute to improvements in the manufacturing process, making it easier and less expensive to produce sustainable materials that aggregate scientific/technological value and can assist in the protection of the environment. The natural dyes used in the present study were selected due to their low toxicity, making them less likely to provoke allergic reactions. The process described in this work led to the even dyeing of BC leather. Further studies are needed to improve the waterproofing process so that the leather can be folded without losing the layer that protects it from moisture. However, the present results prove the potential of reconstituted BC as a vegan alternative to animal leather.

## Figures and Tables

**Figure 1 jfb-13-00049-f001:**
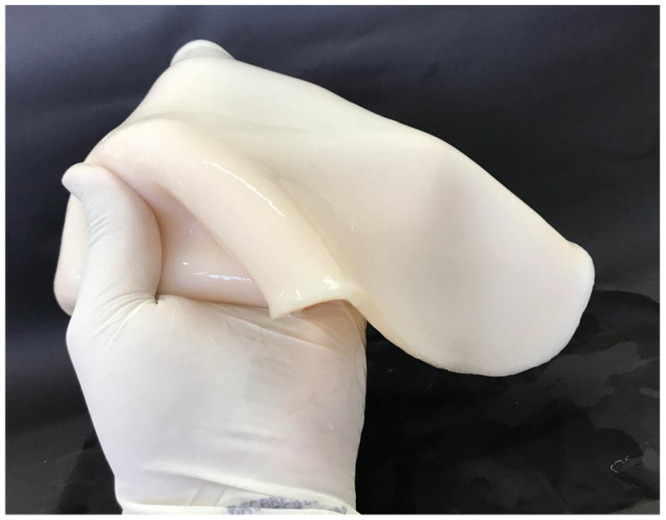
Bacterial cellulose after the purification process.

**Figure 2 jfb-13-00049-f002:**
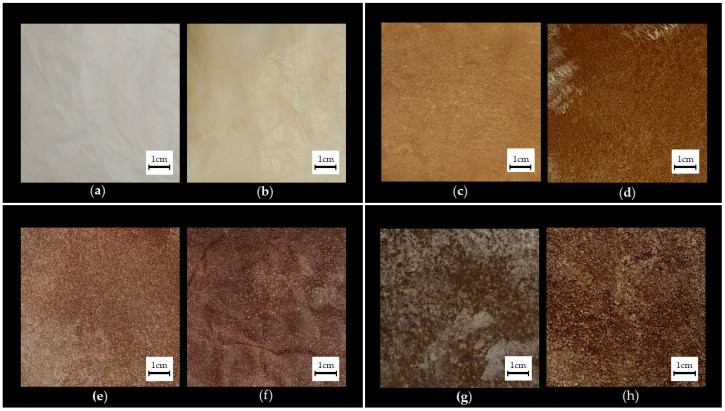
Appearance of cellulose leathers before and after dyeing/processing (**a**) Pure Bacterial Cellulose (BC); (**b**) Waterproofed Bacterial Cellulose (BC-W); (**c**) Reconstituted Bacterial Cellulose (BR); (**d**) Waterproofed Reconstituted Bacterial Cellulose (BR-W); (**e**) Reconstituted Bacterial Cellulose Dyed With Onion and Pomegranate (BRA); (**f**) Waterproofed Reconstituted Bacterial Cellulose Dyed With Onion and Pomegranate (BRA-W); (**g**) Reconstituted Bacterial Cellulose Dyed with Eucalyptus (BRE); (**h**) Waterproofed Reconstituted Bacterial Cellulose Dyed with Eucalyptus (BRE-W).

**Figure 3 jfb-13-00049-f003:**
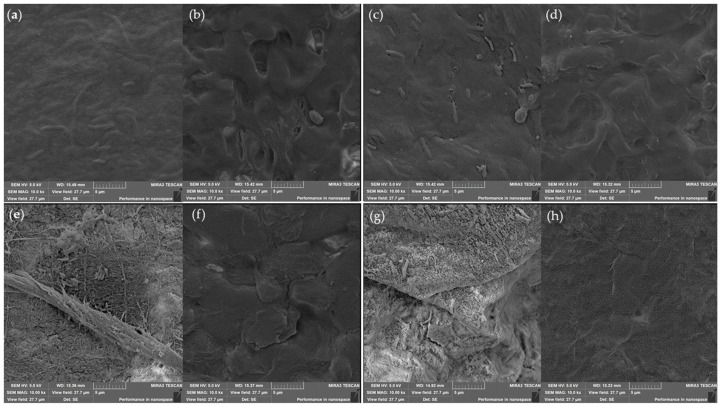
Scanning electron microscopy of all samples: (**a**) Pure Bacterial Cellulose (BC); (**b**) Waterproofed Bacterial Cellulose (BC-W); (**c**) Reconstituted Bacterial Cellulose (BR); (**d**) Waterproofed Reconstituted Bacterial Cellulose (BR-W); (**e**) Reconstituted Bacterial Cellulose Dyed With Onion and Pomegranate (BRA); (**f**) Waterproofed Reconstituted Bacterial Cellulose Dyed With Onion and Pomegranate (BRA-W); (**g**) Reconstituted Bacterial Cellulose Dyed with Eucalyptus (BRE); (**h**) Waterproofed Reconstituted Bacterial Cellulose Dyed with Eucalyptus (BRE-W).

**Figure 4 jfb-13-00049-f004:**
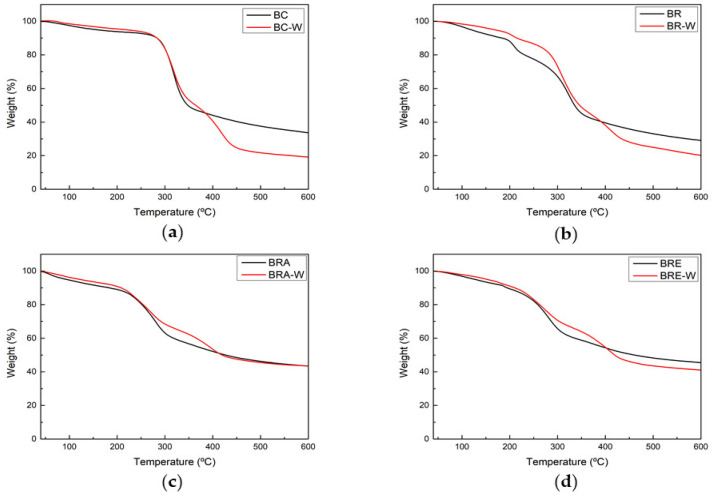
Thermogravimetric graphs of samples: (**a**) Pure (BC) and waterproofed (BC-W) bacterial cellulose; (**b**) Reconstituted (BR) and waterproofed (BR-W) bacterial cellulose; (**c**) Reconstituted (BRA) and waterproofed (BRA-W) bacterial cellulose dyed with onion and pomegranate; (**d**) Reconstituted and (BRE) and waterproofed (BRE-W) bacterial cellulose dyed with eucalyptus.

**Figure 5 jfb-13-00049-f005:**
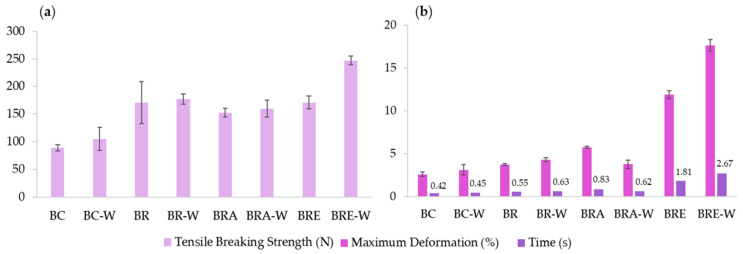
Graphs of results of mechanical tests: (**a**) Tensile strength (N); (**b**) Maximum deformation (%) according to time (s). BC: Pure Bacterial Cellulose; BC-W: Water-proofed bacterial cellulose; BR: Reconstituted bacterial cellulose; BR-W: Water-proofed reconstituted bacterial cellulose; BRA: Reconstituted bacterial cellulose dyed with onion and pomegranate; BRA-W: Water-proofed reconstituted bacterial cellulose dyed with onion and pomegranate; BRE: Reconstituted bacterial cellulose dyed with eucalyptus; BRE-W: Water-proofed reconstituted bacterial cellulose dyed with eucalyptus.

**Table 1 jfb-13-00049-t001:** Bacterial cellulose production yield and water retention capacity.

BC 14 Days	Yield (g/L) Mean ± Standard Deviation	WRC (%) Mean ± Standard Deviation
Wet weight	422.12 ± 15.26	97.62 ± 0.39
Dry weight	10.07 ± 1.97	

**Table 2 jfb-13-00049-t002:** Bacterial cellulose samples and abbreviations.

Sample	Abbreviation
Pure bacterial cellulose	BC
Waterproofed bacterial cellulose	BC-W
Reconstituted bacterial cellulose	BR
Water-proofed reconstituted bacterial cellulose	BR-W
Reconstituted bacterial cellulose dyed with onion and pomegranate	BRA
Water-proofed reconstituted bacterial cellulose dyed with onion and pomegranate	BRA-W
Reconstituted bacterial cellulose dyed with eucalyptus	BRE
Water-proofed reconstituted bacterial cellulose dyed with eucalyptus	BRE-W

**Table 3 jfb-13-00049-t003:** Water contact angle, swelling ratio, and sorption index of bacterial cellulose samples. BC: Pure Bacterial Cellulose; BC-W: Water-proofed bacterial cellulose; BR: Reconstituted bacterial cellulose; BR-W: Water-proofed reconstituted bacterial cellulose; BRA: Reconstituted bacterial cellulose dyed with onion and pomegranate; BRA-W: Water-proofed reconstituted bacterial cellulose dyed with onion and pomegranate BRE: Reconstituted bacterial cellulose dyed with eucalyptus; BRE-W: Water-proofed reconstituted bacterial cellulose dyed with eucalyptus.

Sample	Water Contact Angle (^o^)	Swelling Ratio (%)	Sorption Index (s)
BC	43.26	57.92 ± 3.12	76.32 ± 3.12
BC-W	67.64	3.81± 0.46	600 >
BR	38.58	33.32 ± 4.32	331.22 ± 12.34
BR-W	80.72	16.77 ± 2.33	600 >
BRA	40.60	34.64 ± 1.47	293.32 ± 26.62
BRA-W	76.32	15.85 ± 2.86	600 >
BRE	52.22	31.78 ± 4.74	312.43 ± 17.29
BRE-W	83.96	18.11 ± 1.12	600 >

**Table 4 jfb-13-00049-t004:** Degradation temperatures of bacterial cellulose samples. BC: Pure Bacterial Cellulose; BC-W: Water-proofed bacterial cellulose; BR: Reconstituted bacterial cellulose; BR-W: Water-proofed reconstituted bacterial cellulose; BRA: Reconstituted bacterial cellulose dyed with onion and pomegranate; BRA-W: Water-proofed reconstituted bacterial cellulose dyed with onion and pomegranate; BRE: Reconstituted bacterial cellulose dyed with eucalyptus; BRE-W: Water-proofed reconstituted bacterial cellulose dyed with eucalyptus.

Samples	Stage 1			Stage 2			Stage 3			Mass Loss at 600 °C (%)
	T_max_	T_onset_	T_endset_	T_max_	T_onset_	T_endset_	T_max_	T_onset_	T_endset_	
BC	301.31	94.72	320.84	339.04	322.04	416.30	-	-	-	33.78
BC-W	289.62	91.22	323.01	341.21	325.78	381.59	447.53	390.87	517.92	20.55
BR	283.23	80.05	326.14	349.77	328.91	423.90	-	--	--	29.24
BR-W	269.13	87.59	327.71	342.66	327.71	384.00	440.65	390.98	503.81	20.30
BRA	236.59	83.30	275.16	312.25	275.64	420.52	-	-	--	42.87
BRA-W	224.05	87.08	298.42	369.34	298.90	404.37	429.19	405.21	504.65	43.00
BRE	245.51	84.35	271.90	309.27	272.26	424.96	-	-	-	45.42
BRE-W	216.25	85.22	293.00	372.07	300.47	406.47	437.52	406.68	495.32	40.31

**Table 5 jfb-13-00049-t005:** Tensile strength and maximum deformation as a function of time of bacterial cellulose samples. BC: Pure Bacterial Cellulose; BC-W: Water-proofed bacterial cellulose; BR: Reconstituted bacterial cellulose; BR-W: Water-proofed reconstituted bacterial cellulose; BRA: Reconstituted bacterial cellulose dyed with onion and pomegranate; BRA-W: Water-proofed reconstituted bacterial cellulose dyed with onion and pomegranate; BRE: Reconstituted bacterial cellulose dyed with eucalyptus; BRE-W: Water-proofed reconstituted bacterial cellulose dyed with eucalyptus.

Sample	Tensile Strength (N)	Maximum Deformation (%)	Time (s)
BC	89.04 ± 11.04	2.61 ± 0.51	0.41 ± 0.05
BC-W	105.11 ± 42.02	3.12 ± 1.19	0.45 ± 0.15
BR	171.01 ± 76.11	3.73 ± 0.23	0.55 ± 0.04
BR-W	177.33 ± 18.55	4.27 ± 0.46	0.63 ± 0.04
BRA	152.53 ± 15.52	5.73 ± 0.23	0.83 ± 0.04
BRA-W	160.02 ± 30.85	3.77 ± 0.96	0.62 ± 0.11
BRE	171.07 ± 23.51	11.87 ± 0.93	1.81 ± 0.14
BRE-W	247.21 ± 16.52	17.63 ± 1.38	2.67 ± 0.18

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
