# Peer review of "Design of a Naturally Dyed and Waterproof Biotechnological Leather from Reconstituted Cellulose"

_jfb, 2022, doi:10.3390/jfb13020049_

Round 1
Reviewer 1 Report
I enjoyed reading your work, and I have one comment: A description in your paper of the folding method would be a positive addition. Was an instrument used or was it by hand?
Author Response
Response: Dear reviewer, thank you very much for the question! No instrument was used, the method was carried out by hand. This information was included in the text for better understanding.
Reviewer 2 Report
The manuscript of da Silva Junior describes the applicability of bacterial nanocellulose to prepare material resembling leather, naming it vegan leather. The manuscript is interesting. The authors may consider the following points for improvement:
- Vegan leather is a very general term, including also plastic-based leather. The authors may consider defining this more precisely in the title and the text.
- The methods for thermal stability: what was the limit point to define the material as thermostable or not?
- Please add the information about the source of SCOBY.
- The authors may consider adding the recent review paper on the production and applicability of bacterial cellulose of Gorgieva and Trcek (2019, Nanomaterials).
Author Response
- Vegan leather is a very general term, including also plastic-based leather. The authors may consider defining this more precisely in the title and the text.
Response: The reviewer is quite right. That is a great observation. The term was modified both in the title and text in order to be more precise.
- The methods for thermal stability: what was the limit point to define the material as thermostable or not?
Response: The limit point on the thermal stability depends on the desired application. All bacterial cellulose samples demonstrated a maximum degradation temperature ranging from 309 ºC to 440 ºC, which shows a great thermal stability for the desired apparel application.
- Please add the information about the source of SCOBY.
Response: The source of SCOBY was added. Thank you for the remark.
- The authors may consider adding the recent review paper on the production and applicability of bacterial cellulose of Gorgieva and Trcek (2019, Nanomaterials).
Response: The paper was added, as suggested by the reviewer.
Reviewer 3 Report
- In the abstract section: The authors need not to describe the process, such as, “The dying process involved the use of plant-based natural dyes extracted from Allium cepa L., Punica granatum and Eucalyptus globulus L. The BC films were then shredded and reconstituted to produce uniform surfaces with a constant thickness of 0.10 cm through the entire area. The films were waterproofed using the essential oil from Melaleuca alternifolia and wax from Copernicia prunifera.”
- In the materials section: Please write the source of the materials and the specifications.
- In section 2.2: “the green tea medium” Name the company, specification etc.
- In line 120: “the biomaterial to adsorb and fix dyes”. Is it ‘adsorb’ or ‘absorb’?
- In section 2.3: Provide conditions for measuring the ‘wet weight’. Time, temperature etc.
- In section 2.5: “heated for 30 min” What is the temperature?
- In section 2.6: Please provide the concentration range for this statement “used for dyeing1000 g of BC fibres (wet mass).”
- Section 2.7: “Each dyed BC film was shredded” Is it in wet condition or dry condition?
- In section 2.9.2: “immersed in a 100-mL distilled water bath at 37 °C for 24 h.” Why did the authors choose 37 deg C?
- In Figure 4: Please correct the ligand inside the graph.
- What is the stability of the dye/coloring materials into the polymer matrix? The authors need to provide some ideas.
Author Response
- In the abstract section: The authors need not to describe the process, such as, “The dying process involved the use of plant-based natural dyes extracted from Allium cepa L., Punica granatum and Eucalyptus globulus L. The BC films were then shredded and reconstituted to produce uniform surfaces with a constant thickness of 0.10 cm through the entire area. The films were waterproofed using the essential oil from Melaleuca alternifolia and wax from Copernicia prunifera.”
Response: We appreciate the observation, however, the description of the used processes were included in order to facilitate the reading and understanding of the work. Also, as there is a large variety of methods that can be applied to the BC films, we aimed to facilitate researchers on the specific techniques that we used.
2. In the materials section: Please write the source of the materials and the specifications.
Response: Thank you for the remark. The sources and specifications that were not included previously, are now provided.
3. In section 2.2: “the green tea medium” Name the company, specification etc.
Response: The details of the used materials were added.
4. In line 120: “the biomaterial to adsorb and fix dyes”. Is it ‘adsorb’ or ‘absorb’?
Response: The term is indeed 'adsorb', as according to the characterization tests that were carried out, we can only guarantee that the dyes were fixed on the surface of the films. However, we appreciate the comment and intend to work in the future towards other tests that can guarantee the complete absorption of the dye into the biomaterial.
5. In section 2.3: Provide conditions for measuring the ‘wet weight’. Time, temperature etc.
Response: Thank you for the observation, this information has been added to the text. The membranes were weighed under environmental conditions (room temperature).
6. In section 2.5: “heated for 30 min” What is the temperature?
Response: This information has been added to the text. The heating took place at 90ºC. We appreciate the remark.
7. In section 2.6: Please provide the concentration range for this statement “used for dyeing1000 g of BC fibres (wet mass).”
Response: The concentration of used substrates for dyeing is described in item 2.4.
8. Section 2.7: “Each dyed BC film was shredded” Is it in wet condition or dry condition?
Response: The BC was ground in its wet condition. This information has been added to the text. Thank you.
9. In section 2.9.2: “immersed in a 100-mL distilled water bath at 37 °C for 24 h.” Why did the authors choose 37 deg C?
Response: Thank you for the kind observation, it was a typo, the analyses were performed at room temperature (25ºC). This information has been corrected.
10. In Figure 4: Please correct the ligand inside the graph.
Response: This information was corrected.
11. What is the stability of the dye/coloring materials into the polymer matrix? The authors need to provide some ideas.
Response: Our initial idea was to understand the possibility of the development of this material and its main properties. However, we appreciate your suggestion. This test is interesting, and we will definitely take it into consideration for the continuity of the work.
Reviewer 4 Report
Bacterial cellulose was fabricated and used as vegan leather. Its dyability and other properties were examined in this paper.
- Page 1, Line 43: “The water contact angle was 83.96°, indicating little absorption of water by the biotextile.” It should be noticed that the water contact angle (WCA) and water absorption are two different concepts. WCA can’t indicate the water absorption.
- The properties of vegan leathers should be compared with the real leather.
- How about its washability?
- Page 6, Figure 2: There must be a scale bar for each picture.
- Page 7, Line 259: “The swelling of the tissue offers…” What does the tissue mean?
- Page 8, Figure 3: Any SEM with high magnification to see the BC fibrils with a diameter of 10 to 100 nm?
Author Response
- Page 1, Line 43: “The water contact angle was 83.96°, indicating little absorption of water by the biotextile.” It should be noticed that the water contact angle (WCA) and water absorption are two different concepts. WCA can’t indicate the water absorption.
Response: We appreciate your remark, the information has been corrected.
2. The properties of vegan leathers should be compared with the real leather.
Response: This research is an initial development of a vegan leather based on bacterial cellulose. This comparison is really interesting, we appreciate it and we will consider it in the next step to be developed in the next manuscript.
3. How about its washability?
Response: Our initial idea was to understand the possibility of the development of this material and its main properties, as mentioned in the previous comment. However this test is interesting and we will definitely take it into consideration for the continuity of the work.
4. Page 6, Figure 2: There must be a scale bar for each picture.
Response: Thank you for the observation! The scale bar was added for Figure 2.
5. Page 7, Line 259: “The swelling of the tissue offers…” What does the tissue mean?
Response: We are sorry, it was meant to be "fabric". The term has been corrected in the text. Thanks.
6. Page 8, Figure 3: Any SEM with high magnification to see the BC fibrils with a diameter of 10 to 100 nm?
Response: Our aim for this SEM test was to analyse the surface of the samples in order to understand if the dyeing and waterproofing would change the morphology of the BC membranes. However, we appreciate the observation, and this information will be taken into account for future works.